# Implicit Regularization of SGD via Thermophoresis

## Abstract

A central ingredient in the impressive predictive performance of deep neural networks is optimization via stochastic gradient descent (SGD). While some theoretical progress has been made, the effect of SGD in neural networks is still unclear, especially during the early phase of training. Here we generalize the theory of *thermophoresis* from statistical mechanics and show that there exists an effective force from SGD that pushes to reduce the gradient variance in certain parameter subspaces. We study this effect in detail in a simple two-layer model, where the thermophoretic force functions to decreases the weight norm and activation rate of the units. The strength of this effect is proportional to squared learning rate and inverse batch size, and is more effective during the early phase of training when the model's predictions are poor. Lastly we test our quantitative predictions with experiments on various models and datasets.

## 1 Introduction

Deep neural networks have achieved remarkable success in the past decade on tasks that were out of reach prior to the era of deep learning. Yet fundamental questions remain regarding the strong performance of over-parameterized models and optimization schemes that typically involve only first-order information, such as stochastic gradient descent (SGD) and its variants.

In particular, optimization via SGD is known in many cases to result in models that generalize better than those trained with full-batch optimization. To explain this, much work has focused on how SGD navigates towards so-called flat minima, which tend to generalize better than sharp minima (Hochreiter & Schmidhuber, 1997; Keskar et al., 2017). This has been argued by nonvacuous PAC-Bayes bounds (Dziugaite & Roy, 2017) and Bayesian evidence (Smith & Le, 2018). More recently, Wei & Schwab (2019) discuss how optimization via SGD pushes models to flatter regions within a minimal valley by decreasing the trace of the Hessian.

However, these perspectives apply to models towards the end of training, whereas it is known that proper treatment of hyperparameters during the early phase is vital. In particular, when training a deep network one typically starts with a large learning rate and small batch size if possible. After training has progressed, the learning rate is annealed and decreased so that the model can be further trained to better fit the training set (Krizhevsky et al., 2012; Simonyan & Zisserman, 2015; He et al., 2016b;a; You et al., 2017; Vaswani et al., 2017). Crucially, using a small learning rate during the first phase of training usually leads to poor generalization and also result in large gradient variance practically (Jastrzebski et al., 2020; Faghri et al., 2020).

However, limited theoretical work has been done to understand the effect of SGD on the early phase of training. Jastrzebski et al. (2020) argue for the existence of a "break-even" point on an SGD trajectory. This point depends strongly on the hyperparameter settings. They argue that the break-even point with large learning rate and small batch size tends to have a smaller leading eigenvalue of the Hessian spectrum, and this eigenvalue sets an upper bound for the leading eigenvalue beyond this point. They also present experiments showing that large learning rate SGD will reduce the variance of the gradient. However their analysis focuses only on the leading eigenvalue of the Hessian spectrum and requires the strong assumption that the loss function in the leading eigensubspace is quadratic.

Meanwhile Li et al. (2020) studied the simple setting of two-layer neural networks. They demonstrate that in this model, training with large learning rate in the early phase tends to result in better generalization than training with small learning rate. To explain this, they hypothesize a separation of features in the data: easy-to-generalize yet hard-to-fit features, and hard-to-generalize, easier-to-fit features. They argue that a model trained with small learning rate will memorize easy-to-generalize, hard-to-fit patterns during phase one, and then generalize worse on hard-to-generalize, easier-to-fit patterns, while the opposite scenario occurs when training with large learning rate. However, this work relies heavily on the existence of these two distinct types of features in the data and the specific network architecture. Moreover, their analysis focuses mainly on learning rate instead of the effect of SGD.

In this paper, we study the dynamics of model parameter motion during SGD training by borrowing and generalizing the theory of thermophoresis from physics. With this framework, we show that during SGD optimization, especially during the early phase of training, the activation rate of hidden nodes is reduced as is the growth of parameter weight norm. This effect is proportional to squared learning rate and inverse batch size. Thus, thermophoresis in deep learning acts as an implicit regularization that may improve the model's ability to generalize.

We first give a brief overview of the theory of thermophoresis in physics in the next section. Then we generalize this theory to models beyond physics and derive particle mass flow dynamics microscopically, demonstrating the existence of thermophoresis and its relation to relevant hyperparameters. Then we focus on a simple two-layer model to study the effect of thermophoresis in detail. Notably, we find the thermophoretic force is strongest during the early phase of training. Finally, we test our theoretical predictions with a number of experiments, finding strong agreement with the theory.

## 2 THERMOPHORESIS IN PHYSICS

Thermophoresis, also known as the Soret effect, describes particle mass flow in response to both diffusion and a temperature gradient. The effect was first discovered in electrolyte solutions (Ludwig, 1859; Soret, 1897; Chipman, 1926). However it was discovered in other systems such as gases, colloids, and biological fluids and solid (Janek et al., 2002; Köhler & Morozov, 2016).

Thermophoresis typically refers to particle diffusion in a continuum with a temperature gradient. In one method of analysis, the non-uniform steady-state density $\rho$ is given by the "Soret Equilibrium" (Eastman, 1926; Tyrell & Colledge, 1954; Wurger, 2014),

$$\nabla \rho + \rho S_T \nabla T = 0 \ , \tag{1}$$

where $T$ is temperature and $S_T$ is called the Soret coefficient.

In other work by de Groot & Mazur (1962), mass flow was calculated by non-equilibrium theory. They considered two types of processes for entropy balance: a reversible process stands for the entropy transfer and an irreversible process corresponds to the entropy production, or dissipation. The resulting mass flow induced by diffusion and temperature gradient was found to be

$$J = -D\nabla \rho - \rho D_T \nabla T \ , \tag{2}$$

where $D$ is the Einstein diffusion coefficient and $D_T$ is defined as thermal diffusion coefficient. Comparing the steady state in 1 and setting the flow to be zero, the Soret coefficient is simply

$$S_T = \frac{D_T}{D} \ . \tag{3}$$

The Soret coefficient can be calculated from molecular interaction potentials based on specific molecular models (Wurger, 2014).

## 3 THERMOPHORESIS IN GENERAL

In this section, we first study a kind of generalized random walk that has evolution equations for a particle state with coordinate $\mathbf{q} = \{q_i\}_{i=1,\dots,n}$ as

$$\mathbf{q}_{t+1} = \mathbf{q}_t - \eta \gamma \mathbf{f}(\mathbf{q}_t, \xi) \ , \tag{4}$$

where $\mathbf{f}$ is a vector function, $\gamma$ and $\xi$ are random variables, and $\eta$ is a small number controlling the step size. Notice that this is a generalized inhomogeneous random walk for the particle. Before further analysis, it is noted that the evolution equations 4 is similar to SGD updates in machine learning and we will show this in the next section.

To isolate the effect of thermophoresis, we assume the random walk is unbiased, in which case

$$P(\gamma \mathbf{f}(\mathbf{q}, \xi) = \mathbf{a}) = P(\gamma \mathbf{f}(\mathbf{q}, \xi) = -\mathbf{a}), \tag{5}$$

for an arbitrary vector $\mathbf{a}$. Thus there is no explicit force exerted on the particle. This simplification was used to demonstrate a residual thermophoretic force in the absence of a gradient. Including gradients is straightforward and corresponds to an external field that creates a bias term. We also denote the probability density, which we also call the mass density, as $\rho(\mathbf{q})$ and

$$g_i(\mathbf{q}) := \sqrt{\int \gamma^2 f_i^2(\mathbf{q}, \xi) d\mu(\gamma, \xi)}, \tag{6}$$

so that $\eta g_i(\mathbf{q})$ is the standard deviation of the random walk in the $i$th direction.

From a position $\mathbf{q}$, we consider a subset of coordinate indices, $U \subseteq \{1, \ldots, n\}$, wherein

$$\text{sign}(f_i(\mathbf{q}, x)) = \text{sign}(f_j(\mathbf{q}, x)) \text{ and } \partial_i g_j(\mathbf{q}) \geq 0 \tag{7}$$

for all $i, j \in U$. We note here that indices will correspond to parameters when we study learning dynamics. The first property is necessary for our derivation. The second condition will be used at the end to conclude that each $g_i$ decreases.

In order to study the dynamics of the particle and its density function, we focus on the probability mass flow induced by the inhomogeneous random walk. We will show that there is always a flow from regions with larger $g_i(\mathbf{q})$ to those with smaller $g_i(\mathbf{q})$ for $i \in U$, which is a generalization of thermophoresis in physics.

Since $\eta \ll 1$, the movement of the particle will have a mean free path of $g_i(\mathbf{q})$ in $i$th direction. Therefore the random walk equation 4 becomes

$$q_i = q_i - \eta g_i(\mathbf{q}) \zeta_i, \tag{8}$$

where $i = 1, \ldots, n$ and $\zeta_i$ is a binary random variable with $P(\zeta_i = -1) = P(\zeta_i = 1) = 0.5$. Moreover, from Eq. 7, we also have that $\zeta_i = \zeta_j$ for all $i$ and $j \in U$.

Next we will show that the flow projecting on the subspace $U$ is always toward smaller $g_i(\mathbf{q})$. Notice that although $U$ can be multi-dimensional, the degree of freedom of the particle dynamics is 1 within $U$ due to the sharing of the $\zeta$s, and therefore the mass flow projecting on it is also 1-dimensional. For each $i \in U$, we define the average flow in this dimension to be the mass that enters $q_i$ from $q_i^-$ minus the mass from the opposite direction $q_i^+$. From Eq. 8 and the assumption that $\eta \ll 1$, only mass close to $q_i$ will move across $q_i$ at each step. We let the farthest mass that will flow across $q_i$ in step $i$ be $q_i + \Delta_i^+$ and $q_i - \Delta_i^-$, where $\Delta_i^+$ and $\Delta_i^+$ are positive. $\Delta_i^+$ and $\Delta_i^-$ are thus defined implicitly by the equations: $\Delta_i^+ = \eta g_i(\mathbf{q} + \boldsymbol{\Delta}^+)$ and $\Delta_i^- = \eta g_i(\mathbf{q} - \boldsymbol{\Delta}^-)$, respectively. Notice that if the random walk were homogeneous, we would have $\Delta_i^+ = \Delta_i^-$. In our inhomogeneous case, we have $\Delta_i^+ \sim \Delta_i^- \sim \eta g_i(\mathbf{q})$ up to leading order of $\eta$, and the next to leading order will be calculated in order to compute the difference between $\Delta_i^+$ and $\Delta_i^-$.

Now we are ready to calculate the mass flow through $\mathbf{q}$. The mass flow projecting onto the subspace $U$ is calculated by the mass through $\mathbf{q}$ from $\mathbf{q} + \boldsymbol{\Delta}^+$ minus the mass from $\mathbf{q} - \boldsymbol{\Delta}^-$ where $\Delta_i^+$ and $\Delta_i^-$ are as above for $i \in U$. It is straightforward to show that[1]

$$\Delta_i^+ - \Delta_i^- = 2\eta^2 \sum_{j \in U} g_j(\mathbf{q}) \partial_j g_i(\mathbf{q}) + O(\eta^3). \tag{9}$$

With this, we can compute the flow density, $J$, through $\mathbf{q}$, finding

$$J = -\eta^2 \sqrt{\sum_{i \in U} g_i^2(\mathbf{q})} \sum_{i \in U} g_i(\mathbf{q}) \partial_i \rho(\mathbf{q}) - \eta^2 \frac{\sum_{i,j \in U} g_i(\mathbf{q}) g_j(\mathbf{q}) \partial_j g_i(\mathbf{q})}{\sqrt{\sum_{i \in U} g_i^2(\mathbf{q})}} \rho(\mathbf{q}) + O(\eta^3), \tag{10}$$

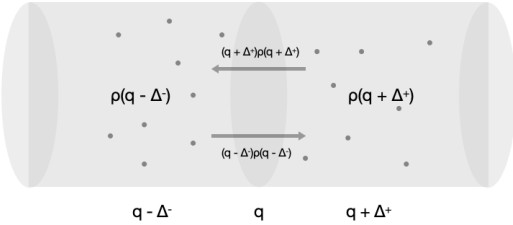

Figure 1: Diagram of mass flow in a generalized inhomogeneous random walk used in the derivation of the Soret coefficient.

where the derivation can be found in Appendix A.3. This can be understood as described in Diagram 1. Notice that this probability mass flow consists of two terms at order $\eta^2$. The first represents diffu-

sion and the second corresponds to our goal in this section, namely the flow due to thermophoresis. By the second property of the $g_i$ in Eq. 7, we find that the coefficient of thermophoresis (Soret coefficient), which is defined as

$$c := -\eta^2 \frac{\sum_{i,j \in U} g_i(\mathbf{q}) g_j(\mathbf{q}) \partial_j g_i(\mathbf{q})}{2 \sqrt{\sum_{i \in U} g_i^2(\mathbf{q})}} \tag{11}$$

$$\leq 0, \tag{12}$$

is negative. This means that there is an effective force exerted on a particle at position $\mathbf{q}$ towards the smaller variance regime (by analogy, the colder area). The coefficient is proportional to $\eta^2$.

## 4 MATHEMATICAL MODEL

### 4.1 TWO-LAYER MODEL

To study the physics behind SGD optimization in detail, we consider the simple setting of one-hidden layer neural networks. The network is a function $f : \mathbb{R}^M \to \mathbb{R}$ parameterized as follows:

$$f(\mathbf{x}; \mathbf{V}, \mathbf{W}, \mathbf{b}) = \mathbf{V}\sigma(\mathbf{W}\mathbf{x} + \mathbf{b}),$$
$$= \sum_{i=1}^{N} V_i \sigma\left(\sum_{j=1}^{M} W_{ij} x_j + b_i\right).$$

We also write $f(x)$ for simplicity. The network has a scalar output, which is widely used in regression and binary classification. $x$ is the network input with dimension $M$, $\mathbf{W}$ and $\mathbf{b}$ are the weights and biases in the first layer with dimension $N \times M$ and $N$ respectively, where $N$ is the number of hidden nodes in the hidden layer, and $\sigma$ is the ReLU activation function defined as $\sigma(a) = \max(0, a)$.

The dataset is drawn i.i.d. from the data distribution, $\{(\mathbf{x}, y)|(\mathbf{x}, y) \sim \mathcal{D}(\mathbf{x}, y)\}$. In this paper we consider two cases, where either $x_i \geq 0$[2] or $x_i \sim \mathcal{N}(0, 1)$[3]. Here $y \in \mathbb{Y}$ and we denote the marginal distribution of $y$ as $\mathcal{D}_Y$. Finally, we have the loss function $L : \mathbb{R} \times \mathbb{Y} \to \mathbb{R}^+$.

### 4.2 TRAINING

We consider optimization via SGD, where the gradient of the loss on a batch of size $|B|$ is given by

---

[1] A brief derivation can be found in Appendix A.2.

[2] Usually in convolutional neural networks or intermediate layers.

[3] Often found when the data are normalized.

$$\nabla \mathcal{L}_B(\mathbf{V}, \mathbf{W}, \mathbf{b}) = \frac{1}{|B|} \sum_{i=1}^{|B|} \nabla_f L(f(\mathbf{x}_i), y_i) \nabla f(\mathbf{x}_i). \tag{13}$$

In our two-layer model, we have

$$\nabla_{V_i} f(\mathbf{x}) = \sigma(\sum_{j=1}^{M} W_{ij} x_j + b_i),$$

$$\nabla_{W_{ij}} f(\mathbf{x}) = V_i x_j \sigma'(\sum_{k=1}^{M} W_{ik} x_k + b_i),$$

$$\nabla_{b_i} f(\mathbf{x}) = V_i \sigma'(\sum_{k=1}^{M} W_{ik} x_k + b_i).$$

For an input vector $\mathbf{x}$, we call the hidden node $i$ *activated* when $\sigma'(\sum_{k=1}^{M} W_{ik} x_k + b_i) = 1$, or equivalently $W_{ik} x_k + b_i > 0$. We thus define the *activation rate* of the network to be

$$\overline{\sigma'} = \frac{1}{N} \sum_{i=1}^{N} \mathbb{E}_x \sigma'(\sum_{k=1}^{M} W_{ik} x_k + b_i) \ . \tag{14}$$

This is an important concept to which we will return.

Henceforth, we drop the index $i$, since the dynamical equations are invariant with respect to node index, and write $V := V_i$, $W_j := W_{ij}$ and $b := b_i$ by abuse of notation. We also denote

$$h_v(V, \mathbf{W}, b) := \mathbb{E}_x[\nabla_V f(\mathbf{x})]^2,$$
$$h_{\mathbf{w}}(V, \mathbf{W}, b) := \mathbb{E}_x[\nabla_{\mathbf{W}} f(\mathbf{x})]^2,$$
$$h_b(V, \mathbf{W}, b) := \mathbb{E}_x[\nabla_b f(\mathbf{x})]^2,$$

where $\mathbb{E}_x$ denotes average over input $\mathbf{x}$. We have the following property for the functions $h$:

**Property 4.1.** *Given $\mathbf{W}$, if $V_1^2 \leq V_2^2$ and $b_1 \leq b_2$, we have*

$$h_v(V_1, \mathbf{W}, b_1) \leq h_v(V_2, \mathbf{W}, b_2),$$
$$h_{\mathbf{w}}(V_1, \mathbf{W}, b_1) \leq h_{\mathbf{w}}(V_2, \mathbf{W}, b_2),$$
$$h_b(V_1, \mathbf{W}, b_1) \leq h_b(V_2, \mathbf{W}, b_2),$$
$$\overline{\sigma'}(V_1, \mathbf{W}, b_1) \leq \overline{\sigma'}(V_2, \mathbf{W}, b_2).$$

*Here we define $\mathbf{a} \leq \mathbf{b}$ as $\min(\mathbf{b} - \mathbf{a}) \geq 0$.*

It is straightforward to see the following:

**Property 4.2.** *When the case of $x_i \geq 0$ is considered, if $V_1^2 \leq V_2^2$, $\mathbf{W}_1 \leq \mathbf{W}_2$ and $b_1 \leq b_2$, we have*

$$h_v(V_1, \mathbf{W}_1, b_1) \leq h_v(V_2, \mathbf{W}_2, b_2),$$
$$h_{\mathbf{w}}(V_1, \mathbf{W}_2, b_1) \leq h_{\mathbf{w}}(V_2, \mathbf{W}, b_2).$$
$$h_b(V_1, \mathbf{W}_2, b_1) \leq h_b(V_2, \mathbf{W}, b_2),$$
$$\overline{\sigma'}(V_1, \mathbf{W}, b_1) \leq \overline{\sigma'}(V_2, \mathbf{W}, b_2).$$

In our analysis, we focus for simplicity on binary classification tasks, where the loss is typically binary cross-entropy: $L(f, y) = y \ln p(f) + (1 - y) \ln(1 - p(f))$ and $p(f) = 1/(1 + \exp(f))$. We thus have

$$\nabla_f L(f, y) = p(f) - y. \tag{15}$$

Substituting into Eq. 13, the mini-batch gradient becomes

$$\nabla \mathcal{L}_B(\mathbf{V}, \mathbf{W}, \mathbf{b}) = \frac{1}{|B|} \sum_{i=1}^{|B|} (p_i - y_i) \nabla f(\mathbf{x}_i). \tag{16}$$

Our results also hold straightforwardly for squared error.

## 5 THERMOPHORESIS IN DEEP LEARNING

In this section, we will show that the parameters in the one hidden layer model and their dynamics approximately satisfy the criteria of the previous section and that the biases are pushed negative and $V^2$ is suppressed during training, the effects of both of which are proportional to squared learning rate $\eta^2$ and inverse batch size $1/|B|$.

The gradient that dominates model training is defined in 16. Because training samples are i.i.d., the variance of the gradient is

$$\text{var}\big[\nabla\mathcal{L}_B(\mathbf{V}, \mathbf{W}, \mathbf{b})\big] = \text{var}\bigg[\frac{1}{|B|}\sum_{i=1}^{|B|}(p_i - y_i)\nabla f(\mathbf{x}_i)\bigg] \ , \tag{17}$$

$$= \frac{1}{|B|}\text{var}\big[(p - y)\nabla f(\mathbf{x})\big] \tag{18}$$

The gradient has two components: $p - y$ corresponding to $\gamma$ in equation 4 and $\nabla f(\mathbf{x})$ corresponding to $\mathbf{f}(\mathbf{q}, \xi)$. We assume that the dataset is unbiased, in which case $P(y = 0) = P(y = 1) = 0.5$ and $P(p - y = a) = P(p - y = -a)$, and that $p - y$ and $\nabla f(\mathbf{x})$ are independent in the first period of training given that the dataset is complex and can't be learned by linear model. It is straightforward to see that it satisfies Eq. 5.

Next we will show that $V$ and $b$ are always in the set of $U$, i.e. they satisfy the conditions of Eq. 7. First, if $V_i \geq 0$, we have

$$\nabla_{V_i} f(\mathbf{x}) = \sigma(\sum_{j=1}^{M} W_{ij}x_j + b_i), \tag{19}$$

$$\geq 0. \tag{20}$$

and

$$\nabla_{b_i} f(\mathbf{x}) = V_i \sigma'(\sum_{k=1}^{M} W_{ik}x_k + b_i), \tag{21}$$

$$\geq 0. \tag{22}$$

Since we also have Property 4.1, the conditions in Eq. 7 are satisfied. If $V_i < 0$, we consider a coordinate transform that maps $V_i$ to $\bar{V}_i = -V_i$. It is easy to show that Eq. 7 is again satisfied after this transform.

Next we consider $\mathbf{W}$. The gradient of $f$ with respect to $W_{ij}$ is the product of $\nabla_{b_i} f$ and $x_i$. If $x_i$ for $i = 1, \ldots, M$ are always $\geq 0$, which is usually the case in convolutional neural networks, it is easy to show that $W_{ij}$ is also in set $U$ and smaller $W_{ij}$ corresponds to smaller variance according to Property 4.2. If $x_i \sim \mathcal{N}(0, 1)$, on the other hand, $\mathbf{W}$ is excluded from $U$.

For the following, we only consider the case where $x_i \sim \mathcal{N}(0, 1)$, and

$$g_V(V_i, \mathbf{W}_i, b_i) = \frac{1}{\sqrt{|B|}}\sqrt{\int\big[(p - y)\sigma(\sum_{j=1}^{M} W_{ij}x_j + b_i)\big]^2 d\mu(x, y)} \ , \tag{23}$$

$$:= \frac{1}{\sqrt{|B|}}\phi_1(\mathbf{W}_i, b_i) \ , \tag{24}$$

$$g_b(V_i, \mathbf{W}_i, b_i) = \frac{1}{\sqrt{|B|}}\sqrt{\int\big[(p - y)V_i\sigma'(\sum_{j=1}^{M} W_{ij}x_j + b_i)\big]^2 d\mu(x, y)} \ , \tag{25}$$

$$:= \frac{V_i}{\sqrt{|B|}}\phi_2(\mathbf{W}_i, b_i) \ , \tag{26}$$

where $g$ is defined as in Eq. 6. Inserting these into Eq. 10, we find the thermophoresis flow density to be

$$J_t = \frac{\eta^2}{|B|}\psi \ , \tag{27}$$

where $\psi = \frac{V_i\phi_1\phi_2^2 + V_i\phi_1\phi_2\partial_b\phi_1 + V^3\phi_2^2\partial_b\phi_2}{2\sqrt{\phi_1^2 + V_i^2\phi_2^2}}\rho$. This flow biases the model toward smaller $b_i$ and smaller $V_i$[4] with a strength proportional to squared learning rate $\eta^2$ and inverse batch size. We also note that $\psi$ can be bounded by a function multiplying with a scalar $\int(p-y)^2\mu(x,y)$. It is clear that this scalar measures the L-2 distance between model predictions and sample labels and decreases on average during training as prediction getting better. Thus thermophoresis is more effective during the early phase of training.

Therefore there exists an effective force that pushes to decrease the model's activation rate, defined in equation 14, and reduces the weight norm of the second layer. The strength of this force scales as

$$F \propto \frac{\eta^2}{|B|} \quad . \tag{28}$$

In Li & Liang (2018), Theorem 4.1 presents a linear relation between learning rate and training iterations for a target training error $\epsilon$ and small learning rate. This implies that if one uses a learning rate $k$ times larger, the model will require $k$ times fewer optimization steps for the same training performance. Together with our results, this implies the following: for the same model and initialization, comparing two optimization schemes with $\eta_1 \leq \eta_2$ each achieving a given training error, the activation rate for scheme 1 will be at least as large as that for scheme 2, i.e. $\sigma_1 \geq \sigma_2$. Similarly, denoting the weight norm for scheme 1(2) by $v_1(v_2)$, we have that $v_1 \geq v_2$.

Model sparsity can mean two different things: sparsity of the weights, and frequency with which units are activated, called the activation rate. Intuitively, a sparser model has a smaller capacity Bizopoulos & Koutsouris (2020); Kurtz et al. (2020); Aghasi et al. (2017); Lin et al. (2017). Furthermore, certain forms of model pruning have been shown to improve generalization Frankle & Carbin (2019); Frankle et al. (2020). Therefore one might surmise that a smaller activation rate in general correlates with generalization. Moreover, in Appendix A.5, we construct an upper bound of the Hessian norm and this upper bound monotonously depends on activation rate and weight norm. This also sheds light on the connection between sparsity, weight norm, and generalization.

Our theory can also be generalized beyond two-layer models. We have shown that there exists an effective force in deep neural networks from SGD that reduces the gradient variance and have derived quantitative properties of it.

## 6 EXPERIMENTS

The essential result from the previous section is that there exists an effective force from SGD, analogous to thermophoresis, that pushes to decrease the gradient variance, and in one-hidden-layer neural networks decreases the model's activation rate and reduces the weight norm of the second layer. The strength of the force is proportional to squared learning rate and inverse batch size. In this section, we present experiments to test these results. Further experiments can be found in the appendix.

First we consider a one hidden layer model with input dimension 100 and 100 hidden units. The input data, $\mathbf{x}$, is distributed as $\mathcal{N}(0, \mathrm{I})$ where I is identity matrix, and the label is randomly chosen from $\{0, 1\}$. Batch size is set to 1 and the learning rate is varied from 0.025 to 0.1. We calculate the activation rate and L2 norm of the vector $\mathbf{V}$ after each training iteration. The result for activation rate is shown in the first row of Fig. 2. The leftmost plot shows activation rate as a function of true iteration on the x-axis, and we see that activation rate decreases during training, and the decreasing is more rapid with larger learning rate. In the middle plot we rescale the x-axis by a factor proportional to learning rate $\eta$[5]. This rescaling factor is to offset the movement difference due to learning rate difference. It is clear that even after this rescaling, we still observe that larger learning rates decrease activation rate faster. Finally, on the rightmost plot we rescale the x-axis with a factor proportional to squared learning rate $\eta^2$. We see that all trajectories now overlap, which matches our prediction in the previous section that decreasing rate is proportional to $\eta^2$.

---

[4]larger $V_i$ if $V_i < 0$.

[5]For example, if raw iteration number for $\eta = 0.05$ is 1000 and rescaled iteration number is also 1000, the rescaled iteration number for $\eta = 0.1$ is 1000 then its true iteration number is 500.

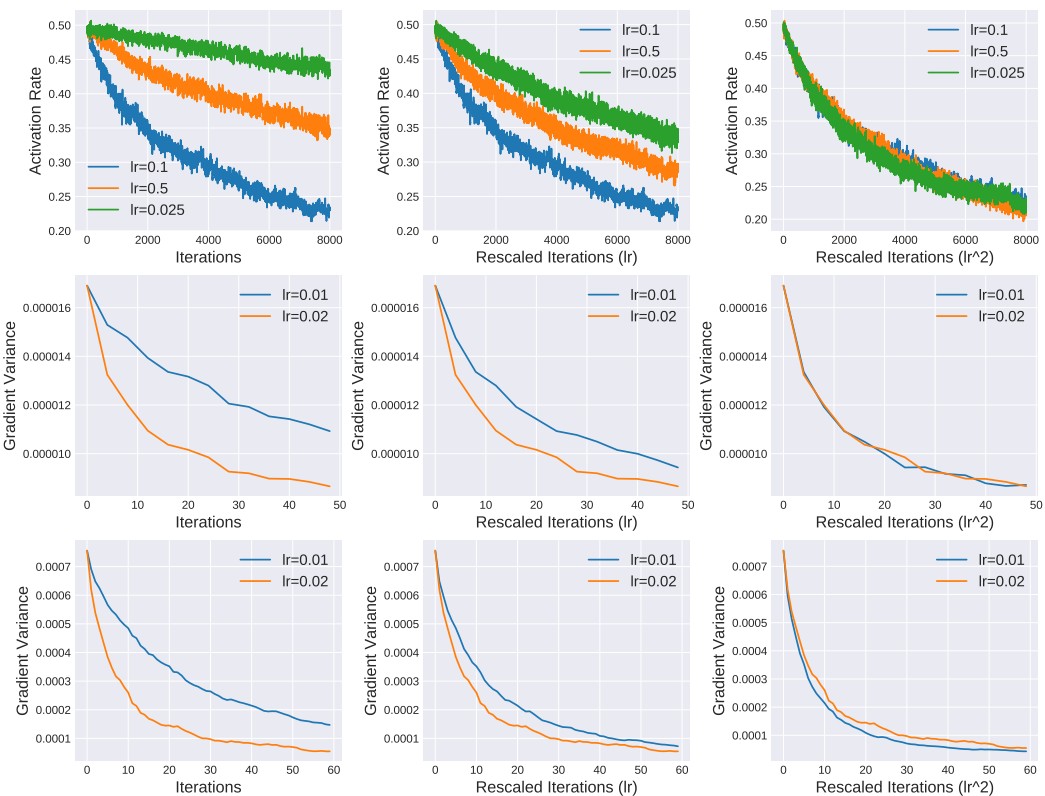

Figure 2: All rows include rescaled x-axes as described in the main text. Top Row: Plots of activation rate as a function of (rescaled) training iterations with different learning rates. The model is a two-layer fully-connected network with 100 hidden units. Training data is drawn from a normal distribution. Middle Row: Plots of average gradient variance as a function of (rescaled) training iterations with different learning rates in 6-layer fully-connected neural networks. Training data is drawn from normal distribution. Bottom Row: Same as middle row but for 6-layer convolutional neural networks trained on Fashion-MNIST.

We next test our results for deep neural networks beyond the two-layer model. Instead of activation rate and weight norm, we plot the gradient variance as predicted by our theory. Networks architectures are 6-layer fully-connected with hidden layer sizes of 100 and 6-layer convolutional with 10 channels with kernel size of 5*5 and stride 1 except the last fully-connected layer output. The results are shown in the second row of Fig. 2 and the third row of Fig. 2 respectively.

Further experiments can be found in Appendix A.6, where we show that the evolution of the weight norm, scaling with batch size, and other results are consistent with our theoretical predictions. We also study other models and other datasets including CIFAR10.

## 7 CONCLUSION

In this paper we generalized the theory of *thermophoresis* from statistical mechanics and showed that there exists an effective thermophoretic force from SGD that pushes to reduce the gradient variance. We studied this effect in detail for a simple two-layer model, where the thermophoretic force serves to decrease the weight norm and the activation rate of the units. We found that the strength of this effect is proportional to square of the learning rate, inversely proportional to batch size, and is more effective during the early phase of training when the model's predictions are poor. We found good agreement between our predictions and experiments on various models and datasets.

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

# A  APPENDIX

## A.1  PROOF OF PROPERTY 4.1

*Proof.* By definition, we have

$$\sigma(\sum_{j=1}^{M} W_j x_j + b_1) \leq \sigma(\sum_{j=1}^{M} W_j x_j + b_2) \ , \tag{29}$$

$$\sigma'(\sum_{j=1}^{M} W_j x_j + b_1) \leq \sigma'(\sum_{j=1}^{M} W_j x_j + b_2) \ . \tag{30}$$

For $h_v$,

$$h_v(V_1, \mathbf{W}, b_1) = \mathbb{E}_x g_v^2(\mathbf{x}, V_1, \mathbf{W}, b_1) \ ,$$

$$= \mathbb{E}_x \sigma^2(\sum_{j=1}^{M} W_j x_j + b_1) \ ,$$

$$\leq \mathbb{E}_x \sigma^2(\sum_{j=1}^{M} W_j x_j + b_2) \ ,$$

$$= h_v(V_2, \mathbf{W}, b_2) \ .$$

Similarly, we have

$$h_{w_i}(V_1, \mathbf{W}, b_1) = \mathbb{E}_x V_1^2 x_i^2 \sigma'(\sum_{k=1}^{M} W_k x_k + b_1) \ ,$$

$$\leq \mathbb{E}_x V_2^2 x_i^2 \sigma'(\sum_{k=1}^{M} W_k x_k + b_1) \ ,$$

$$\leq \mathbb{E}_x V_2^2 x_i^2 \sigma'(\sum_{k=1}^{M} W_k x_k + b_2) \ ,$$

$$= h_{w_i}(V_2, \mathbf{W}, b_2) \ .$$

Clearly the inequality also holds for $h_b$. □

## A.2  DERIVATION OF EQ. 9

$$\Delta_i^+ - \Delta_i^- = \eta g_i(\mathbf{q} + \mathbf{\Delta}^+) - \eta g_i(\mathbf{q} - \mathbf{\Delta}^-) \ , \tag{31}$$

$$= \eta \sum_{j \in U}^{n} (\Delta_j^+ + \Delta_j^-) \partial_j g_i(\mathbf{q}) + O(\eta \Delta^2) \ , \tag{32}$$

$$= 2\eta^2 \sum_{j \in U} g_j(\mathbf{q}) \partial_j g_i(\mathbf{q}) + O(\eta^3) \ . \tag{33}$$

## A.3  DERIVATION OF EQ. 10

$$
\begin{aligned}
J &= -\frac{1}{2}|\mathbf{\Delta}^+|\rho(\mathbf{q} + \mathbf{\Delta}^+) + \frac{1}{2}|\mathbf{\Delta}^-|\rho(\mathbf{q} - \mathbf{\Delta}^-) \ , \\
&= \frac{1}{2}|\mathbf{\Delta}^+|[\rho(\mathbf{q} - \mathbf{\Delta}^-) - \rho(\mathbf{q} + \mathbf{\Delta}^+)] + \frac{1}{2}(|\mathbf{\Delta}^-| - |\mathbf{\Delta}^+|)\rho(\mathbf{q} - \mathbf{\Delta}^-) \ , \\
&= -\frac{1}{2}|\mathbf{\Delta}^+|(|\mathbf{\Delta}^+ + \mathbf{\Delta}^-|)\frac{\rho(\mathbf{q} + \mathbf{\Delta}^+) - \rho(\mathbf{q} - \mathbf{\Delta}^-)}{|\mathbf{\Delta}^+ + \mathbf{\Delta}^-|} \\
&\quad -\frac{1}{2}\frac{(|\mathbf{\Delta}^+|^2 - |\mathbf{\Delta}^-|^2)}{|\mathbf{\Delta}^+| + |\mathbf{\Delta}^-|}\rho(\mathbf{q} - \mathbf{\Delta}^-) \ , \\
&\approx -\frac{1}{2}|\mathbf{\Delta}^+|(\mathbf{\Delta}^+ + \mathbf{\Delta}^-)\nabla\rho(\mathbf{q}) - \frac{1}{2}\frac{\sum_{i\in U}(\Delta_i^+ + \Delta_i^-)(\Delta_i^+ - \Delta_i^-)}{|\mathbf{\Delta}^+| + |\mathbf{\Delta}^-|}\rho(\mathbf{q} - \mathbf{\Delta}^-) \ , \\
&= -\eta^2\sqrt{\sum_{i\in U} g_i^2(\mathbf{q})}\sum_{i\in U} g_i(\mathbf{q})\partial_i\rho(\mathbf{q}) - \eta^2\frac{\sum_{i,j\in U} g_i(\mathbf{q})g_j(\mathbf{q})\partial_j g_i(\mathbf{q})}{\sqrt{\sum_{i\in U} g_i^2(\mathbf{q})}}\rho(\mathbf{q}) + O(\eta^3) \ .
\end{aligned}
$$

## A.4  SANITY CHECK OF GENERALIZED THEORY

If $|U| = 1$ and $g_i(\mathbf{q}) = g_i(q_i)$, the model will reduce to aforementioned physics model and the Soret coefficient reduces to

$$
c = \frac{\eta^2}{2}g(q)g'(q) \ , \tag{34}
$$

$$
= [(\frac{\eta g(q)}{2})^2]' \ , \tag{35}
$$

$$
\approx \nabla T \ , \tag{36}
$$

where $T$ is the effective temperature in the model. This result is consistent with thermophoresis model in physics.

## A.5  SPARSITY, WEIGHT NORM AND THEIR RELATION TO GENERALIZATION

In this section, we demonstrate how sparsity is related to the Hessian norm. We first denote the model's probabilistic prediction on a $C$-class classification as

$$
p_k^\mu = \frac{\exp z_k^\mu}{\sum_{l=1}^{C}\exp z_l^\mu} \ , \tag{37}
$$

where $k$ is the probability for label $k$, $\mu$ is the data index, $z$ is model output and $C$ is the total number of categories. We consider cross entropy loss of the form

$$
L(w) = -\frac{1}{B}\sum_{\mu=1}^{B}\sum_{k=1}^{C} y_k^\mu \log p_k^\mu \ , \tag{38}
$$

where $y$ is sample labels and $p$ stands for model probability prediction, similar to the previous definition. We denote the loss for individual sample to be

$$
L^\mu = -\sum_{k=1}^{C} y_k^\mu \log p_k^\mu \ . \tag{39}
$$

The gradient with respect to the model output is

$$
(\nabla_z L^\mu)_k = -y_k^\mu + p_k^\mu \ . \tag{40}
$$

And it is easy to show that the Hessian with respect to output is

$$
(\nabla_z^2 L^\mu)_{kl} = \delta_{kl}p_k^\mu - p_k^\mu p_l^\mu \ . \tag{41}
$$

Therefore the Hessian with respect to model parameters is

$$H^\mu = \nabla_w^2 L(z(w)) \ , \tag{42}$$
$$= \nabla_w(\nabla_z L * \nabla_w z) \ , \tag{43}$$
$$= (\nabla_w z)(\nabla_z^2 L)(\nabla_w z) + \nabla_z L \nabla_w^2 z \ , \tag{44}$$
$$\approx (\nabla_w z^\mu)_{ij}(\nabla_z^2 L^\mu)_{jk}(\nabla_w z^\mu)_{kl} \ . \tag{45}$$

To study the spectrum of the Hessian, we calculate the trace and have

$$Tr(H^\mu) \approx Tr((\nabla_w z^\mu)(\nabla_z^2 L^\mu)(\nabla_w z^\mu)^T) \ , \tag{46}$$
$$= Tr((\nabla_z^2 L^\mu)(\nabla_w z^\mu)^T(\nabla_w z^\mu)) \ , \tag{47}$$
$$= Tr(P * K) \ , \tag{48}$$
$$\leq Tr(P) * Tr(K) \ , \tag{49}$$

where

$$P = \nabla_z^2 L^\mu \ , \tag{50}$$
$$K_{\mu\nu} = \sum_l \sum_{ij} (\frac{\partial z_\mu}{\partial w_{ij}^l})(\frac{\partial z_\nu}{\partial w_{ij}^l}) \ . \tag{51}$$

The trace of K therefore can be calculated by chain rule,

$$Tr(K) = \sum_\mu \sum_l \sum_{ij} (\frac{\partial z_\mu}{\partial w_{ij}^l})^2 \ , \tag{52}$$
$$= \sum_l (\sum_{iu} (\delta_i^l[\mu])^2 \sum_j (h_j^{l-1})^2) \ , \tag{53}$$

where $\delta$ and $h$ carry backward and forward information respectively. They are defined as

$$\delta_i^l[\mu] = \delta_{\mu n_L} W_{n_L n_{L-1}}^L \sigma'...W_{n_{l+2}i}^{l+1}\sigma' \ , \tag{54}$$
$$h_j^{l-1} = \sigma W_{jn_l}^{l-1}\sigma...W_{n_1 n_0}x_{n_0} \ . \tag{55}$$

It can further be shown that

$$\sum_{iu} (\delta_i^l[\mu])^2 = \sum_{iu} \delta_{\mu n_L} W_{n_L n_{L-1}}^L \sigma' \ldots W_{n_{l+2}i}^{l+1}\sigma' * \sigma' \overline{W^{l+1}}_{in_{l+2}} \ldots \sigma' \overline{W^L}_{n_{L-1}n_L} \delta_{n_L\mu} \ , \tag{56}$$

$$= Tr(W^L \sigma' \ldots W^{l+1}\sigma'\sigma'\overline{W^{l+1}} \ldots \sigma'\bar{W}^L) \ , \tag{57}$$

$$\leq Tr(\sigma'\overline{W^L}W^L\sigma')Tr(\sigma'\overline{W^{L-1}}W^{L-1}\sigma') \ldots Tr(\sigma'\overline{W^{l+1}}W^{l+1}\sigma') \ , \tag{58}$$

as well as

$$\sum_j (h_j^{l-1})^2 \leq \|X\|_2 Tr(\overline{W^1}\sigma\sigma W^1)Tr(\overline{W^2}\sigma\sigma W^2)...Tr(\overline{W^{l-1}}\sigma\sigma W^{l-1}) \ . \tag{59}$$

Together with the previous calculations and the definition of $K$, we have

$$Tr(K) \leq \|X\|_2 \sum_l \Pi_{n=1}^{l-1} Tr(\overline{W^n}\sigma\sigma W^n)\Pi_{n=l+1}^L Tr(\sigma'\overline{W^n}W^n\sigma') \ , \tag{60}$$

$$= \|X\|_2 \sum_l \Pi_{n=1}^{l-1}\|\sigma W^n\|_F^2 \Pi_{n=l+1}^L \|W^n\sigma'\|_F^2 \ . \tag{61}$$

Finally, we derive an upper bound for the trace of the Hessian,

$$Tr(H^\mu) \leq Tr(P)\|X\|_2 \sum_l \Pi_{n=1}^{l-1}\|\sigma W^n\|_F^2 \Pi_{n=l+1}^L \|W^n\sigma'\|_F^2 \ . \tag{62}$$

Notice that activation rate and weight norm control the magnitude of $\|\sigma W^n\|_F^2$ and $\|W^n\sigma'\|_F^2$. Therefore smaller activation rate and weight norm lead to tiger upper bound of the Hessian trace and thus indicate smaller matrix norm. This analysis connects sparsity with Hessian norm, Hessian trace specifically.

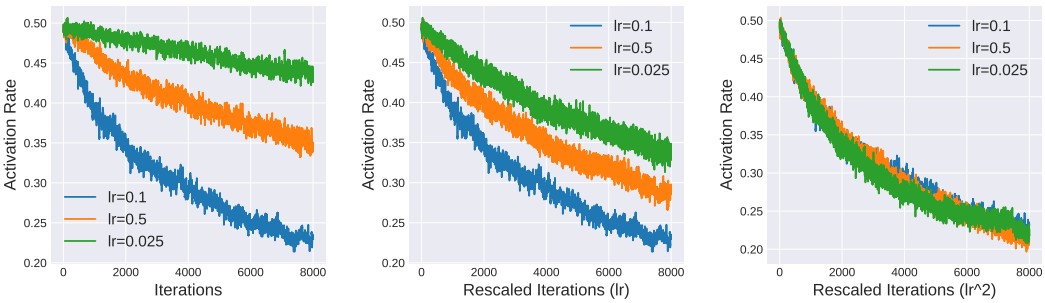

Figure 3: Plots of activation rate as functions of training iterations with different learning rate.

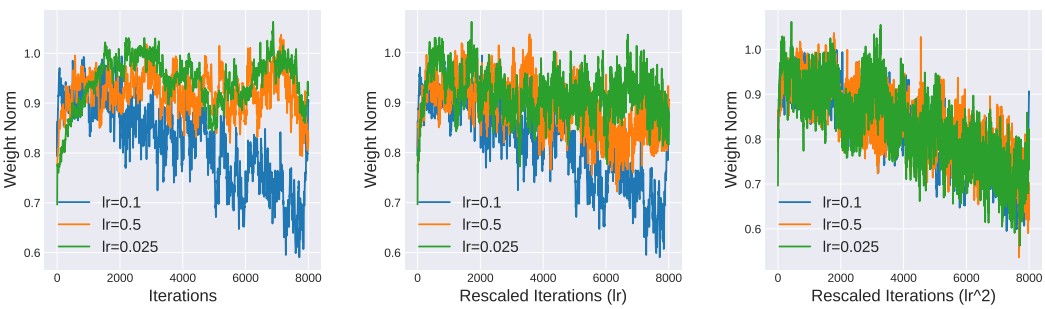

Figure 4: Plots of L2 norm of $\mathbf{V}$ as functions of training iterations with different learning rate.

## A.6 MORE EXPERIMENTS

In this section, we design extensive experiments to test the obtained results. First of all, we consider binary classification with BCE loss. The model has the form

$$f(\mathbf{x}) = \mathbf{V}\sigma(\mathbf{Wx} + \mathbf{b}) \ . \tag{63}$$

Input dimension is $10$ and the number of hidden nodes is also $10$. $\mathbf{x}$ is distributed as $\mathcal{N}(0, \mathrm{I})$ where I is identity matrix and the label is randomly chosen from $\{0, 1\}$.

For the first hyperparameter setting, batch size is $1$ and learning rate varies from $0.025$ to $0.1$. We calculate the activation rate and L2 norm of the vector $\mathbf{V}$ after each training iteration. The result for activation rate and weight norm are shown in Fig. 3 and Fig. 4 respectively. Both figures contain three plots. The leftmost plots correspond to plot with raw iteration as x-axis. It is shown that both activation rate and weight norm are decreasing for all cases. Additionally, the decreasing rate is larger with larger learning rate. The middle plots correspond to plot with rescaled iteration as x-axis, where the rescaled factor is proportional to learning rate $\eta$[6]. This rescalation factor is to offset the movement difference due to learning rate difference. It is clear that even after this rescalation, we still observe activation rate and weight norm difference for different learning rates. Lastly, the rightmost plots correspond to plot with rescaled iteration as x-axis, where the rescaled factor is proportional to squared learning rate $\eta^2$. The result that all trajectories overlap with each other matches our prediction in the previous section, that decreasing rate is proportional to $\eta^2$.

For the second hyperparameter setting, learning rate is fixed to be $0.05$ and batch size varies from $1$ to $3$. Again we calculate the activation rate and L2 norm of the vector $\mathbf{V}$ after each training

---

[6]For example, if raw iteration number for $\eta = 0.05$ is 1000 and rescaled iteration number is also 1000, the rescaled iteration number for $\eta = 0.1$ is 1000 then its true iteration number is 500.

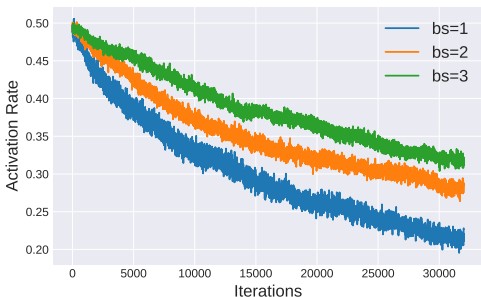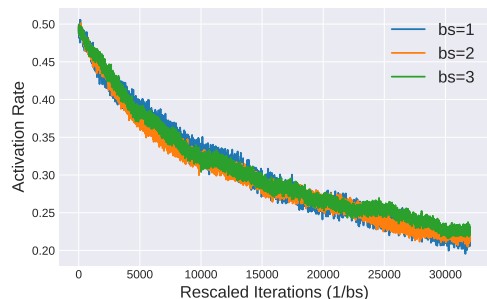

Figure 5: Plots of activation rate as functions of training iterations with different batch size.

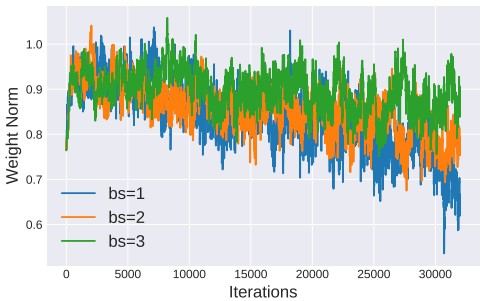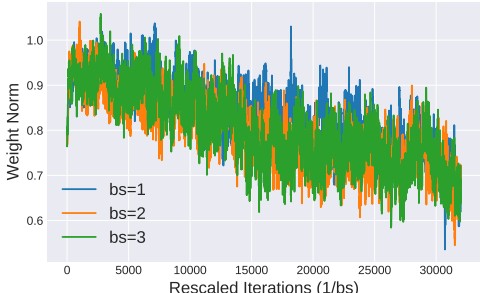

Figure 6: Plots of L2 norm of $\mathbf{V}$ as functions of training iterations with different batch size.

iteration. The result for activation rate and weight norm are shown in Fig. 5 and Fig. 6 respectively. We observe similar tendency as we discussed in the previous hyperparameter setting. The result shows that both activation rate and weight norm are decreasing for all cases. While there exist decreasing rate difference in the left plots due to batch size discrepancy. This difference can be offset be rescaling x-axis according to proportional factor $1/|B|$, which is the result in the right plots and it is consistent with our theory prediction.

Subsequently we consider that second case discussed in the previous section. This is also a consider binary classification with BCE loss. The model, however, has the form

$$f(\mathbf{x}) = \mathbf{V}\sigma(\mathbf{W}\mathbf{x}) \ . \tag{64}$$

Input dimension is $10$ and the number of hidden nodes is also $10$. $\mathbf{x}$ is distributed uniformly between $0$ and $1$ as $U(0,1)$ and the label is randomly chosen from $\{0,1\}$.

The first setting is the same as the first setting in the aforementioned experiment. The result for activation rate and weight norm are shown in Fig. 7 and Fig. 8 respectively. Similar to previous experiment, both figures contain three plots. The leftmost, middle, right plots correspond to plot with raw iteration, rescaled iteration with factor $\eta$ and rescaled iteration with factor $\eta^2$ respectively as x-axis. The result that all trajectories overlap with each other matches our prediction in the previous section, that decreasing rate is proportional to $\eta^2$.

The second setting is also similar to the second setting in the previous experiment where we fix learning rate to be $0.05$ and vary batch size from $1$ to $3$. The result for activation rate and weight norm are shown in Fig. 9 and Fig. 10 respectively. The result shows that both activation rate and weight norm are decreasing for all cases. While there exist decreasing rate difference in the left plots due to batch size discrepancy. This difference can be offset be rescaling x-axis according to

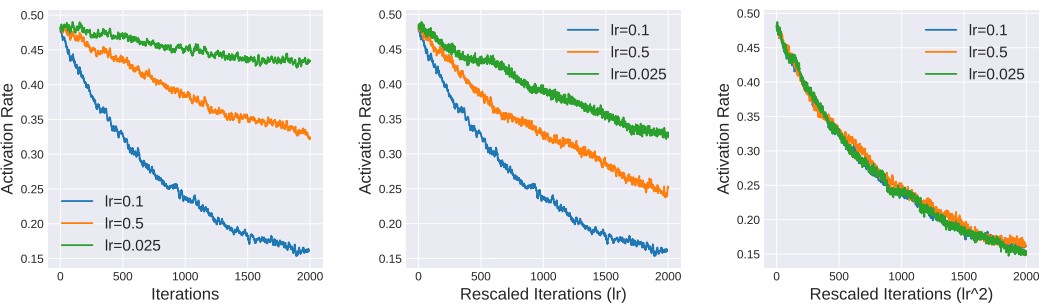

Figure 7: Plots of activation rate as functions of training iterations with different learning rate.

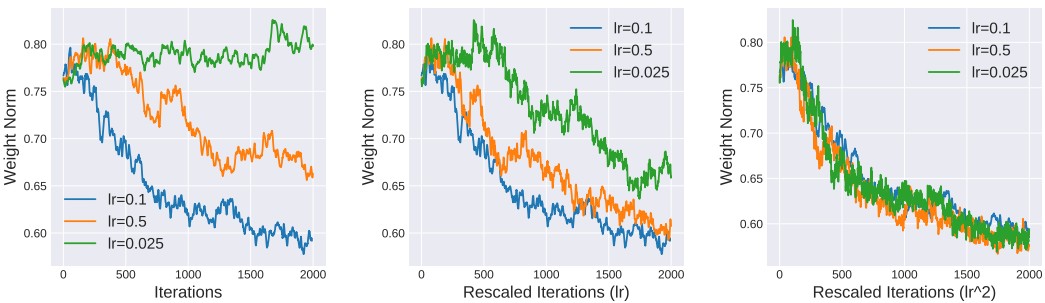

Figure 8: Plots of L2 norm of **V** as functions of training iterations with different learning rate.

proportional factor $1/|B|$, which is the result in the right plots and it is consistent with our theory prediction.

Furthermore, we use real image dataset CIFAR10 for the next experiment instead of artificial data. The model in this experiment has one hidden layer with 300 hidden nodes. We first fix batch size to be 1000 and vary learning rate $\eta$. The result of activation rate and weight norm decreasing are shown in Fig. 11 and Fig. 12 respectively. We then fix learning rate to be 0.02 and vary batch size. The result of activation rate and weight norm decreasing are shown in Fig. 13 and Fig. 14 respectively.

It is clear that the result matches our theoretical predictions. We skip detailed analysis here as it is similar to our discussion in previous experiments.

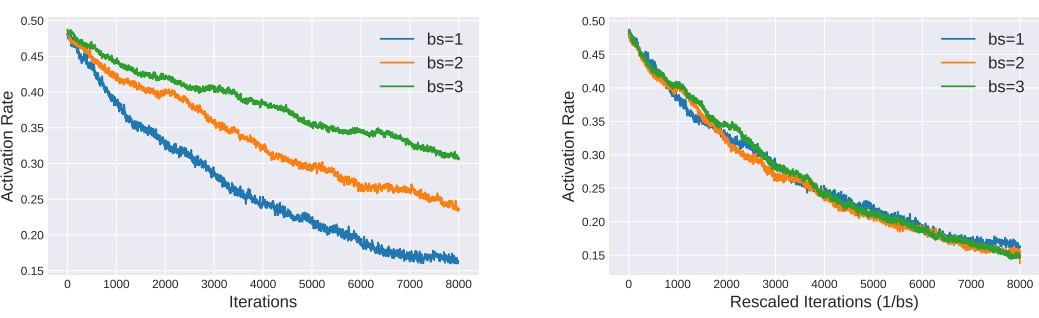

Figure 9: Plots of activation rate as functions of training iterations with different batch size.

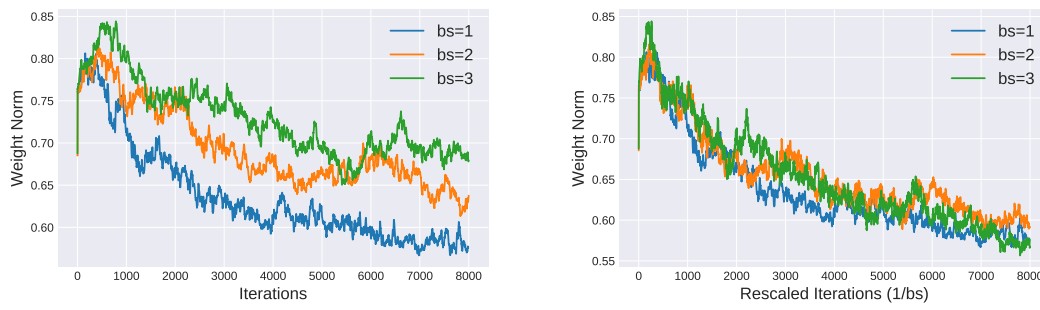

Figure 10: Plots of L2 norm of **V** as functions of training iterations with different batch size.

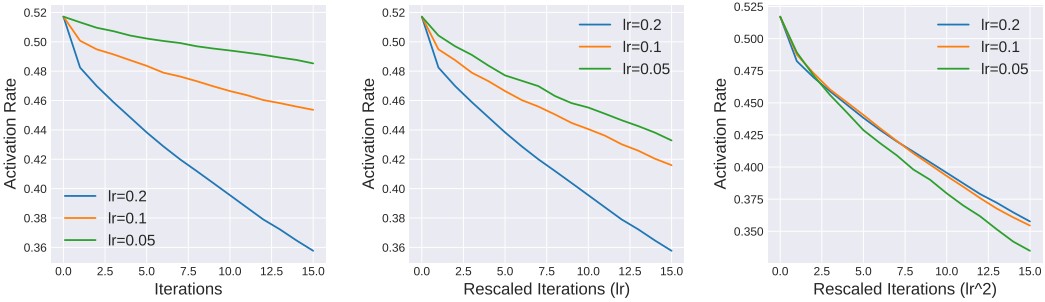

Figure 11: Plots of activation rate as functions of training iterations with different learning rate. Dataset is CIFAR10.

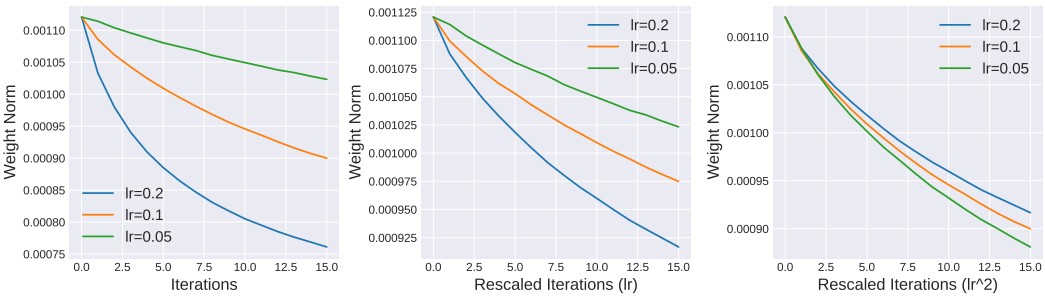

Figure 12: Plots of L2 norm of **V** as functions of training iterations with different learning rate. Dataset is CIFAR10.

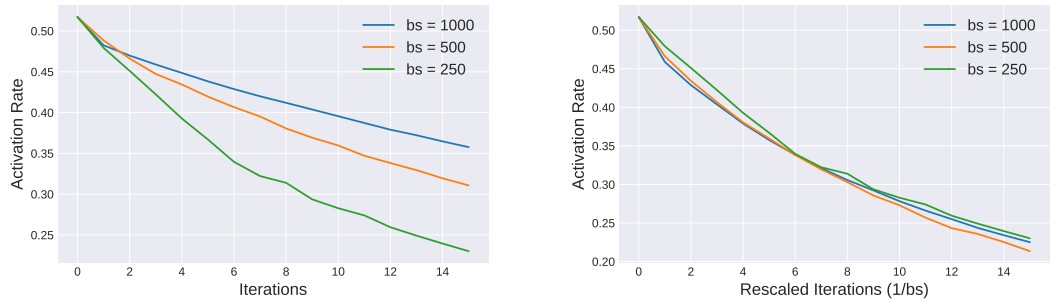

Figure 13: Plots of activation rate as functions of training iterations with different batch size. Dataset is CIFAR10.

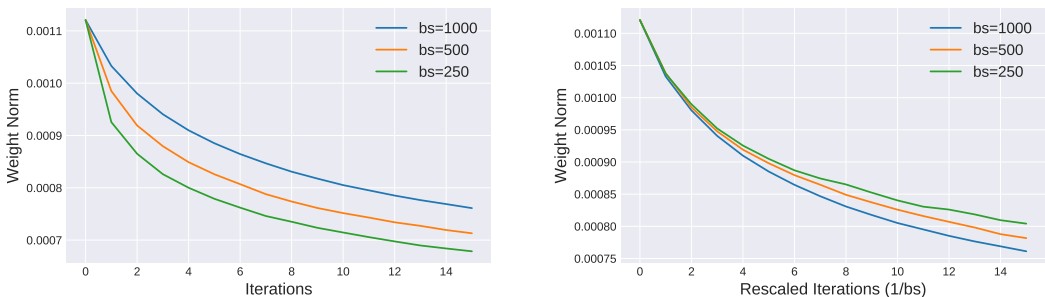

Figure 14: Plots of L2 norm of **V** as functions of training iterations with different batch size. Dataset is CIFAR10.

