# OpenReview forum: "Implicit Regularization of SGD via Thermophoresis"
_ICLR.cc/2021/Conference — Reject_

### Official Review · AnonReviewer3 · 2020-10-27
**Not Fully Clear about the Mathematics / Physics, Not Ideal Empirical Results in Real Datasets**

**Rating:** 3
**Confidence:** 2

**Review:**

This paper studies the implicit regularization effect of SGD from the Thermophoresis perspective. The authors find several quantities scale with square of learning rate over batch size, including activation rate and gradient variance. The theory, to my understanding, requires several strong assumptions, e.g., (1) zero true gradient, and (2) certain implicit independence condition. The empirical results in real dataset like CIFRAR-10 do not seem to support the conclusion, either. As someone from CS/Applied Math community, I personally encourage the authors to write the theory in a more mathematical manner: I find several of the current reasoning in the main text hard to grasp ---- maybe I am missing important Physics backgrounds.

Below are my detailed reviews ---- it is highly possible that I misunderstand something; if so please do clarify.

# Theory

- Contradicting statements in Page 2 the second paragraph vs. Page 2 Eq. (3). In specific you study steady state and let J to be zero. However when the dynamic reaches its steady state, it is no longer directly related to the "early phase", which is claimed as the focused object in Page 2 the second paragraph.

- Eq. (5). I am super confused here. Does Eq. (5) imply the expectation of the gradient is zero??? This is not at all true for GD/SGD. I mean how can you effectively optimize an objective with zero (expected) gradient?

- Eq. (9) and Eq. (10). I cannot understand the paragraphs for Eq. (9) and (10). Recall the definition of U in Eq. (7), I believe U should also depend on time t? Then what do you mean by projecting the flow on to the subspace U, where the subspace U is not even fixed? I tried to look at the appendix, but sorry I am unable to follow the abbreviated derivation.

- Property 4.2. What is W here?

- Below Eq. (18). Why P(p-y = a) = P(p-y = -a)??? To my understand, if we write down the complete formula, it reads
P ( p(f(x_i, theta)) - y_i = a ) = P( p(f(x_i, theta)) - y_i = -a ),
where theta is (W, V, b) the parameters. Even the dataset is unbiased, i.e., P(y = 0) = P(y=1), after theta being injected, we no longer have independence for p, thus I do not understand the above equality.


- Below Eq. (22). Why can you consider a transformation that maps V to -V??? Intuitively this transformation changes a descent iterate to an ascent iterate and vice versa. Does not it change the behavior of the original dynamic??



# Experiments
The results in toy datasets seem to be good. But when I look at Appendix A.6, the plots for CIFAR-10 do not seem to agree with your theory in my humble perspective.

# Vague Statements
- Page 7, the third paragraph. I understand sparsity of weights implies small capacity; however I do not see why sparsity of the activations also imply small capacity. In specific, it can be that the rate of activation is small, but for each data the activated neurons are different. In this case can you prune the network as stated in the paragraph?
- The title and abstract emphasize the paper concerns SGD. But I fail to find a direct connection between SGD and the so called "mass flow" in Eq. (1). I suspect the tile and abstract are over claimed.

Overall, I cannot give high scores to this paper. Again, I can miss important facts because of lacking Physics background.

---

> ### Author Response · Authors · 2020-11-25
> **Response to Reviewer 3**
>
> Thank you for your efforts to understand our work. We hope that with clarification on a number of points you will understand and appreciate our work better. We respond to each of your concerns below.
>
> Theory
>
> "Contradicting statements in Page 2..."
>
> In the second paragraph, we introduce the phenomenon of thermophoresis in physics in order to provide intuition and a better understanding of our later theory. The specifics of this subsection, e.g. steady state dynamics, are not relied upon in our work. In our theory, we study non-equilibrium dynamics of thermophoresis in deep learning with non-zero J as in Eq. 10. We will streamline this background material to address this confusion. Thanks for helping us improve the presentation.
>
> -----
>
> "Eq. (5). I am super confused here."
>
> That is correct: we simplified the setting here to an unbiased random walk in order to focus purely on the entropic effect of SGD. The derivation can be easily generalized to include a bias term in the flow Eq.10, where we will have an extra term corresponding to external field (gradient descent). This generalization will not change the result in this paper. We will update Appendix to include a discussion of this general case.
>
> -------
>
> "Eq. (9) and Eq. (10)."
>
> The set U is not time dependent and only depends on the network architecture and dataset. We consider the requirement for elements in U to be satisfied for arbitrary state and input data from the dataset. For example, in our two-layer case, if the input data is positive, the set U is {V, W, b} throughout all of training, but if the input is Gaussian, the set U is {V, b} in which case the sign of the gradient with respect to W depends on input x and does not satisfy Eq. 7.
>
> -------
>
> "Property 4.2. What is W here?"
>
> W here is the weight matrix of the first layer defined at the beginning of Section 4.1.
>
> ------
>
> "Below Eq. (18)..."
>
> Similar to your second concern, this was chosen to simplify our theory in order to isolate the entropic effect of SGD due to noise; the removal of this setting will result in an extra external field term in Eq. 10, which does not change the result in this paper.
>
> -------
>
> "Below Eq. (22)..."
>
> The transformation we mean is defining a new parameter \bar{V} = -V which will then reduce to our previous analysis. Notice that this is not changing the parameter value in the network and will not change the descent iterate.
> The update rule for \bar{V} can be obtained from
> V = V - dL/dV --> V = V + dL/d(-V) --> (-V) = (-V) - dL/d(-V) -->  \bar{V} =  \bar{V} - dL/d\bar{V}
> The descent iterate is invariant.
>
> -------
>
> Experiments
>
> "The results in toy datasets..."
>
> We would argue that the CIFAR-10 results are in fairly good agreement with our theory. While not as perfect a “data collapse” as the other experiments, they show that quantities such as activation rate and weight norm (a) decrease during training, and (b) the rate of decrease is very nearly--if not exactly--proportional to (learning rate)^2 / (batch size). For (a), it was not a priori obvious that activation rate should decrease during training. For (b), we emphasize that our theory has no free parameters. The curves need not be rescalings of each other at all. If we were to “fit” the scaling of these curves, they would be e.g. ~(learning rate)^1.9, and certainly a far cry from being simply proportional to learning rate as one might naively expect.
>
> -----
>
> Vague Statements
>
> "Page 7, the third paragraph."
>
> This is a great point--it may be that the activation rate is reduced because neurons are disabled for all inputs, which would allow for pruning, or that a different subset of neurons is activated for each input. We are currently running experiments to determine which effect dominates the reduction in activation rate, but we make no claim that reduced activation rate from SGD allows for pruning, and have removed ambiguous statements to that effect. We simply meant to use the beneficial effects of pruning on generalization as an example of sparsity aiding generalization. A stronger point is made in Appendix A.5, where we relate properties of the Hessian that are known to correlate with generalization to the activation rate. In addition to the papers cited in that paragraph, another example in unsupervised learning is sparse coding [1].
> [1] Olshausen BA, Field DJ. Emergence of simple-cell receptive field properties by learning a sparse code for natural images. Nature. 1996 Jun;381(6583):607-9.
>
> --------
>
> "The title and abstract..."
>
> Our paper is indeed studying SGD. “Mass flow” was physics jargon to mean probability mass flow that we have now been careful to define. Apologies for the confusion. We have tried to qualify our claims in particular with regard to the subset U, but are unsure in what way you are saying we overclaim. We would be eager to correct any places where we overclaim.

---

### Official Review · AnonReviewer1 · 2020-10-28

**Rating:** 7
**Confidence:** 3

**Review:**

1. Summary:

This paper proposes that SGD has the implicitly bias of reducing gradient variance via the phenomenon of thermophoresis that masses tend to flow from regions with higher temperature / variance of random walk to regions with lower temperature / variance of random walk. In the setup of two-layer neural networks trained by SGD for binary classification, the authors show the analogous phenomenon that the model is biased towards smaller activation rate and the norm of its second layer weight. The dependence of the rate at which this phenomenon happens on the learning rate and batch size are verified in the experiments.

2. Clearly state the recommendation (accept or reject) with one or two key reasons for this choice:

I am recommending an accept to this paper, as this work presents a reasonable and empirically-supported theory for the implicit regularization effect of SGD on two-layer neural networks.

3. Arguments for the recommendation and questions for the authors:

The authors did a good job explaining the general theory of thermophoresis as well as the derivation of the effect on the two-layer neural network model. The experiment agrees with the theoretical prediction quite nicely.

4. Additional questions and comments:

a) On the bottom of Page 5, the authors claim that the results can be generalized to arbitrary loss. Is this obvious?

b) The condition that $x_i$ is nonnegative seems unnatural in practice, because I think in practice techniques like data whitening are sometimes applied during preprocessing, which centers the input data.

c) On Page 6, the authors mention “the set of U defined in the previous section”, but I couldn’t find it defined in Section 4, and also the set U defined in Section 3 seems like something separate to me - it is a set of indices rather than a set of weights.

d) In the derivation on Page 3, how do we know that $\Delta_i^+ = \Delta_i^- = 0$ if $i \notin U$? Because in the end, which indices belong to $U$ is not fixed, and the complement of a $U$ may also be a legitimate $U$ itself, for example.

---

> ### Author Response · Authors · 2020-11-25
> **Response to Reviewer 1**
>
> Thanks for your positive review! We respond to each of your questions/comments below.
>
>
> "a) On the bottom of Page 5, the authors claim that the results can be generalized to arbitrary loss. Is this obvious?"
>
> Thank you for catching this. Our current analysis only applies to cross entropy and mean squared error--we will carefully reword our claim in the revised version.
>
> ---------
>
> "b) The condition that xi is nonnegative seems unnatural in practice, because I think in practice techniques like data whitening are sometimes applied during preprocessing, which centers the input data."
>
> This is indeed correct. This is why we also consider other types of inputs, in particular Gaussian, in our paper. Moreover, if we consider an intermediate layer of a ReLU network, the input of such layer is indeed nonnegative (the output of a ReLU nonlinearity), although this layer is jointly trained with other layers which is beyond the scope of our paper.
>
> -------
>
> "c) On Page 6, the authors mention “the set of U defined in the previous section”, but I couldn’t find it defined in Section 4, and also the set U defined in Section 3 seems like something separate to me - it is a set of indices rather than a set of weights."
>
> This pointer is actually a typo -- the definition is in Section 3. The indices in Section 3 do indeed correspond to parameters. That is, the indices are coordinate indices and we treat the weights/biases in neural nets as coordinates of neural nets at state q. We will include a clear explanation of this in the revised version.
>
> -------
>
> "d) In the derivation on Page 3, how do we know that Δi+=Δi−=0 if i∉U? Because in the end, which indices belong to U is not fixed, and the complement of a U may also be a legitimate U itself, for example."
>
> We actually don’t require that Δi+=Δi−=0, nor do we use this. Our statement to this effect in the paragraph above Eq. 9 was unnecessary. We apologize for this and will remove this statement. Thank you for the careful reading!

---

### Official Review · AnonReviewer4 · 2020-10-30
**Interesting finding, but the mathematical reasoning is imprecise**

**Rating:** 4
**Confidence:** 3

**Review:**

## Summary

This paper uses the concept of thermophoresis from statistical mechanics to study certain aspects of the dynamics of SGD in neural network training. It is argued that this explains SGD’s tendency to reduce the gradient variance and that the strength of this effect depends on the setting of the step size and batch size.


## Rating

The paper presents an interesting finding, namely, that SGD naturally seeks to reduce gradient variance. This is an interesting potential avenue to understand to effectiveness of SGD in neural network training. The main argument of the paper relies on a physics analogy. For a theoretical paper studying a mathematical process (SGD), the mathematical reasoning seems very vague and I have problems following it in various steps. I am listing some specific questions below. If the mathematical reasoning can be made precise and is explained clearly, this could be a good contribution. In its current form, I don’t think the paper plausibly support its claim of showing “that there exists an effective entropic force from SGD that pushes to reduce the gradient variance”, even for the simple model problem of a single-hidden layer neural net. I therefore recommend **rejection** for now, but would like to encourage the authors to respond to my questions below during the rebuttal phase.


## Individual Comments and Questions

1) The main argument is based on a physics analogy. I put a lot of effort into it, but I just can’t follow the chain of reasoning here. Up to Eq. (8), we are talking about a particle state $q$, which evolves stochastically in discrete time steps. That is a vector-valued discrete time stochastic process. In the paragraph after Eq. (8), the paper is talking about a notion of “mass flow through $q$”. What is that supposed to mean mathematically? It seems to me that we are talking about the probability of transitioning into state $q$ from any other state. Is that correct? I’m all for intuition, but for a theoretical paper, it still needs to be clear what mathematical quantities we are talking about.

2) Eq. (7) restricts consideration to a subset of coordinates. This restriction is not discussed in the paper at all.
    a) Why is that necessary? Is it a technicality or a can we really only say something about such a subset of coordinates.
    b) Is there an intuitive meaning to that subset? The second condition in Eq. (7) seems mysterious and it is not checked whether this holds in the model problem of Section 5.
    c) In the derivation of Eq. (9), why can the “mass flow” be restricted to that set $U$?
    d) How and why can we generalize a finding that holds only on such subset? The abstract characterizes the finding as: “show that there exists an effective entropic force from SGD that pushes to reduce the gradient variance.” Is it really this general?
    e) What would these subsets look like in practical SGD? I can see from Section 5, that such subsets might correspond to layers when using ReLU nonlinearities. But beyond that and when using nonlinearities that can take both positive and negative values, I can’t see where we would have sets of parameters, whose partial derivatives have the same sign **for all inputs $x$**.

3) Glossing over the issues mentioned in point (1), the derivation of the main result relies on **a lot** of relatively crude approximations.
    a) In Eq. (5), an unbiased random walk is assumed. Transferred to SGD, this would imply that the mean gradient is zero, which seems almost fundamentally at odds with the setting of gradient-based optimization. How can this be reconciled. Can a non-zero mean gradient be inserted into the model and can its effect be isolated? This is totally non-obvious to me.
    b) Eq. (5) further implies a symmetric noise distribution, which is a very strong assumption. In the 2-layer NN studied in Section 5, some gradients are strictly positive irrespective of the input $x$ (e.g. Eq. 19), which is basically irreconcilable with the assumption of symmetric gradient noise.
    c) Eq. (8) is a drastic simplification of the true process given by Eq. (4). It is approximating a continuous with a binary increment.
Approximations are necessary when studying complex phenomena. But they need to be carefully discussed and justified. I feel that the paper is not delivering that.

4) The experiments compare gradient variance (and related quantities) along trajectories obtained with different step sizes. It is observed that the per-step reduction in gradient variance per step is roughly proportional to the squared step size, which echos the theoretical findings. But how can we know that this reduction in gradient variance is not just an effect of the gradients becoming smaller overall? It would be much more interesting to see the gradient variance relative to the (squared) norm of the mean gradient as a quantity that characterizes “training progress” independent of the step size. For similar values of the gradient magnitude, does large-step SGD seek out regions with smaller gradient variance then small-step SGD?


## Typos / Style

- Second-to last paragraph of page 1: “Jastrzebski at al. (2020) argues” should be “argue”
- Near bottom of page 1: “large learning SGD” should probably be “large learning rate SGD”
- First paragraph of page 2: “than” should be “then”
- Middle of page 3 says there is a flow “toward negative $g_i(q)$”. I think what you want to say here is that there’s a flow toward **smaller** $g_i(q)$, since $g_i(q)$ is a nonnegative quantity.
- Middle of page 3: $\eta \ll 0$ should be $\eta \ll 1$


## Update after Rebuttal

Thanks for the detailed reply. You have answered a number of my questions, but many of my main concerns remain, in particular points 3 b) and c) as well as 4). Furthermore, incorporating all the clarifications and additional discussions into the paper would in my opinion amount to a substantial revision beyond the usual scope of revisions after rebuttal. I will thus stick to my original rating and recommend rejection for this paper.

---

> ### Author Response · Authors · 2020-11-25
> **Response to Reviewer 4**
>
> Thanks very much for your thoughtful reading, helpful comments, and the opportunity to clarify a number of important points. We're glad you found the direction of the work interesting and hope you will be satisfied with our clarifications of the analysis. We respond to each of your concerns in turn below.
>
> "1. The main argument is based on a physics analogy...."
>
> We would like to emphasize that the physics background in Section 2 is meant only to provide a better intuitive understanding of the theory in Section 3. Our main argument is not based on a physics analogy. Our work, however, does generalize the theory of thermophoresis, so we believe the framing is appropriate.
>
> You are correct--’mass’ flow is indeed a probability flow. We will clarify this and define this terminology properly in the revised paper. We apologize for using physics jargon without first defining this.
>
> ----------
>
> "2. Eq. (7) restricts consideration to a subset of coordinates..."
>
> This question is generally about our restriction to considering a subset of parameters U. We hope to clarify the reason for this restriction and to emphasize that our example in Section 5 demonstrates that it includes realistic cases. We will also qualify all statements in the paper similar to that quoted above so as not to overclaim. Thank you for pointing this out.
>
> a, b) This subset U is constructed to satisfy two conditions. The first condition in Eq. (7) makes it such that if we know the movement direction of one coordinate, the rest can be determined. In our example in Eq.19 and Eq.20, for arbitrary position and subspace (V, b), we can always infer the V (or b) direction if we know how b (or V) changes. Our derivation relies on this condition, and we would be very interested if you saw a way to relax it. The second condition in Eq. (7) is only used at the last step to conclude that variance in the U subspace is reduced. Thus, we can actually conclude when this condition is violated, variance can in fact increase with time. We have indeed checked that the required properties hold in Section 5, from above Eq.(19) to above Eq.(23), relying on Properties 4.1 and 4.2.
>
> c) We mean projection to the subspace U, not that we are restricting the flow. We apologize for the confusion and will clarify this in the revised paper.
>
> d) Indeed our result only holds for subsets of parameters that satisfy the conditions that define U. However we find that this restriction is in fact satisfied for relevant and realistic parameter sets in neural nets. For example, if the input is positive, we see in Section 5 that it contains all the parameters of the two layer model, including weights and biases.
>
> e) Our theory focuses only on ReLU nonlinearities in this paper, and the analysis of networks of arbitrary nonlinearities is beyond the scope of this paper. We believe that ReLU nonlinearities are sufficiently common that our work is still of broad value.
>
> ----------
>
> "3. Glossing over the issues mentioned in point (1)..."
>
> We simplified the problem setting to an unbiased random walk in order to isolate the entropic effect of SGD due to noise. This result can be easily generalized to arbitrary random walk with a bias term in the flow Ep. 10, where we will have an extra term corresponding to external field (gradient descent). This generalization will not change the results in the paper. We will update this extra generalization and discussion in Appendix.
>
> Regarding Eq. (8), we believe this step is justified. One can think of this as temporal coarse graining. In the regime of small eta, many steps will be taken with nearly identical values of the parameters. Thus, the dynamics will be well approximated by Eq. (8), and it is not a drastic simplification.
>
> ---------
>
> "4. The experiments compare gradient variance..."
> In most of our experiments, we measure activation rate and weight norm where this shouldn't be a concern. Our theory shows that motion in the subspace U, and the sqrt of the variance, is proportional to eta^2. The y-axis in Fig. 2 should replace Gradient Variance with g. We apologize for this confusion.
>
> ----------
>
> Thanks for the careful reading, all typos and style issues you found have been be fixed.

---

### Decision · Program_Chairs · 2021-01-07
**Final Decision**

**Decision:**

Reject

**Comment:**

This paper uses concepts from physics to make predictions about stochastic gradient descent. The reviews point to two issues. Firstly, the paper was not very accessible to those without a relevant background, and this is reflected in the low confidence rating reviewers gave. More importantly, two of the reviewers consistently pointed out 'vague mathematics' and oversimplification in the mathematical arguments.

The authors' feedback did not successfully address the reviewer's concerns, both R3 and R4 indicated there were outstanding concerns.

I should note that despite giving low confidence scores and stating that some concepts from physics are beyond their field of expertise, reviewers gave high quality reviews with detailed comments and questions, and subsequently participated in the discussion revisiting their reviews. This suggests that the low confidence is not a symptom of insufficient reviewer effort, but perhaps a consequence of an inaccessible paper.